# Deep Online Learning via Meta-Learning: Continual Adaptation for Model-Based RL

**Anusha Nagabandi, Chelsea Finn & Sergey Levine**
University of California, Berkeley
{nagaban2,cbfinn,svlevine}@berkeley.edu

## Abstract

Humans and animals can learn complex predictive models that allow them to accurately and reliably reason about real-world phenomena, and they can adapt such models extremely quickly in the face of unexpected changes. Deep neural network models allow us to represent very complex functions, but lack this capacity for rapid online adaptation. The goal in this paper is to develop a method for continual online learning from an incoming stream of data, using deep neural network models. We formulate an online learning procedure that uses stochastic gradient descent to update model parameters, and an expectation maximization algorithm with a Chinese restaurant process prior to develop and maintain a mixture of models to handle non-stationary task distributions. This allows for all models to be adapted as necessary, with new models instantiated for task changes and old models recalled when previously seen tasks are encountered again. Furthermore, we observe that meta-learning can be used to meta-train a model such that this direct online adaptation with SGD is effective, which is otherwise not the case for large function approximators. In this work, we apply our meta-learning for online learning (MOLe) approach to model-based reinforcement learning, where adapting the predictive model is critical for control; we demonstrate that MOLe outperforms alternative prior methods, and enables effective continuous adaptation in non-stationary task distributions such as varying terrains, motor failures, and unexpected disturbances. Videos available at: https://sites.google.com/berkeley.edu/onlineviameta

## 1 Introduction

Human and animal learning is characterized not just by a capacity to acquire complex skills, but also the ability to adapt rapidly when those skills must be carried out under new or changing conditions. For example, animals can quickly adapt to walking and running on different surfaces (Herman, 2017) and humans can easily modulate force during reaching movements in the presence of unexpected perturbations (Flanagan & Wing, 1993). Furthermore, these experiences are remembered, and can be recalled to adapt more quickly when similar disturbances occur in the future (Doyon & Benali, 2005). Since learning entirely new models on such short time-scales is impractical, we can devise algorithms that explicitly train models to adapt quickly from small amounts of data. Such online adaptation is crucial for intelligent systems operating in the real world, where changing factors and unexpected perturbations are the norm. In this paper, we propose an algorithm for fast and continuous online learning that utilizes deep neural network models to build and maintain a task distribution, allowing for the natural development of both generalization as well as task specialization.

Our working example is continuous adaptation in the model-based reinforcement learning setting, though our approach generally addresses any online learning scenario with streaming data. We assume that each "trial" consists of multiple tasks, and that the delineation between the tasks is not provided explicitly to the learner – instead, the method must adaptively decide what "tasks" even represent, when to instantiate new tasks, and when to continue updating old ones. For example, a robot running over changing terrain might need to handle uphill and downhill slopes, and might choose to maintain separate models that become specialized to each slope, adapting to each one in turn based on the currently inferred surface.

We perform adaptation simply by using online stochastic gradient descent (SGD) on the model parameters, while maintaining a mixture model over model parameters for different tasks. The mixture is updated via the Chinese restaurant process (Stimberg et al., 2012), which enables new tasks to be instantiated as needed over the course of a trial. Although online learning is perhaps one of the oldest applications of SGD (Bottou, 1998), modern parametric models such as deep neural networks are exceedingly difficult to train online with this method. They typically require medium-sized minibatches and multiple epochs to arrive at sensible solutions, which is not suitable when receiving data in an online streaming setting. One of our key observations is that meta-learning can be used to learn a prior initialization for the parameters that makes such direct online adaptation feasible, with only a handful of gradient steps. The meta-training procedure we use is based on model-agnostic meta-learning (MAML) (Finn et al., 2017), where a prior weight initialization is learned for a model so as to optimize improvement on any task from a meta-training task distribution after a small number of gradient steps.

Meta-learning with MAML has previously been extended to model-based RL (Nagabandi et al., 2018), but only for the $k$-shot adaptation setting: The meta-learned prior model is adapted to the $k$ most recent time steps, but the adaptation is not carried forward in time (i.e., adaptation is always performed from the prior itself). This rigid batch-mode setting is restrictive in an online learning setup and is insufficient for tasks that are further outside of the training distribution. A more natural formulation is one where the model receives a continuous stream of data and must adapt online to a potentially non-stationary task distribution. This requires both fast adaptation and the ability to recall prior tasks, as well as an effective adaptation strategy to interpolate as needed between the two.

The primary contribution of this paper is a meta-learning for online learning (MOLe) algorithm that uses expectation maximization, in conjunction with a Chinese restaurant process prior on the task distribution, to learn mixtures of neural network models that are each updated with online SGD. In contrast to prior multi-task and meta-learning methods, our method's online assignment of soft task probabilities allows for task specialization to emerge naturally, without requiring task delineations to be specified in advance. We evaluate MOLe in the context of model-based RL on a suite of challenging simulated robotic tasks including disturbances, environmental changes, and simulated motor failures. Our simulated experiments show a half-cheetah agent and a hexapedal crawler robot performing continuous model adaptation in an online setting. Our results show online instantiation of new tasks, the ability to adapt to out-of-distribution tasks, and the ability to recognize and revert back to prior tasks. Additionally, we demonstrate that MOLe outperforms a state-of-the-art prior method that does $k$-shot model-based meta-RL, as well as natural baselines such as continuous gradient updates for adaptation and online learning without meta-training.

## 2 RELATED WORK

Online learning is one of the oldest subfields of machine learning (Bottou, 1998; Jafari et al., 2001). Prior algorithms have used online gradient updates (Duchi et al., 2011) and probabilistic filtering formulations (Murphy, 2002; Hoffman et al., 2010; Broderick et al., 2013). In principle, commonly used gradient-based learning methods, such as SGD, can easily be used as online learning algorithms (Bottou, 1998). In practice, their performance with deep neural network function approximators is limited (Sahoo et al., 2017): such high-dimensional models must be trained with batch-mode methods, minibatches, and multiple passes over the data. We aim to lift this restriction by using model-agnostic meta-learning (MAML) to explicitly pretrain a model that enables fast adaptation, which we then use for continuous online adaptation via an expectation maximization algorithm with a Chinese restaurant process (Blei et al., 2003) prior for dynamic allocation of new tasks in a non-stationary task distribution.

Online learning is related to that of continual or lifelong learning (Thrun, 1998), where the agent faces a non-stationary distribution of tasks over time. However, unlike works that focus on avoiding negative transfer, i.e. catastrophic forgetting (Kirkpatrick et al., 2017; Rebuffi et al., 2017; Zenke et al., 2017; Lopez-Paz et al., 2017; Nguyen et al., 2017), online learning focuses on the ability to rapidly learn and adapt in the presence of non-stationarity. While some continual learning works consider the problem of forward transfer, e.g. Rusu et al. (2016); Aljundi et al. (2017); Wang et al. (2017), these works and others in continual learning generally focus on small sets of tasks where

fast, online learning is not realistically possible, since there are simply not enough tasks to recover structure that enables fast, few-shot learning in new tasks or environments.

Our approach builds on techniques for meta-learning or learning-to-learn (Thrun & Pratt, 1998; Schmidhuber, 1987; Bengio et al., 1992; Naik & Mammone, 1992). However, most recent meta-learning work considers a setting where one task is learned at a time, often from a single batch of data (Santoro et al., 2016; Ravi & Larochelle, 2017; Munkhdalai & Yu, 2017; Wang et al., 2016; Duan et al., 2016). In our work, we specifically address non-stationary task distributions and do not assume that task boundaries are known. Concurrent work (Anonymous, 2019) has used a mixture model at the level of the prior parameters, considering non-stationarity at the meta-level, whereas we use a mixture model to model the task-specific parameters to perform online learning only during run-time. Other meta-learning works have considered non-stationarity within a task (Al-Shedivat et al., 2017) and episodes involving multiple tasks at meta-test time (Ritter et al., 2018), but they do not consider continual online adaptation with unknown task separation. Prior work has also studied meta-learning for model-based RL (Nagabandi et al., 2018). This prior method updates the model every time step, but each update is a batch-mode $k$-shot update, using exactly $k$ prior transitions and resetting the model at each step. This allows for adaptive control, but does not enable continual online adaptation, since updates from previous steps are always discarded. In our comparisons, we find that our approach substantially outperforms this prior method. To our knowledge, our work is the first to apply meta-learning to learn streaming online updates.

## 3    PROBLEM STATEMENT

We formalize our online learning problem setting as follows: at each time step, the model receives an input $\mathbf{x}_t$ and produces a prediction $\hat{\mathbf{y}}_t$. It then receives a ground truth label $\mathbf{y}_t$, which must be used to adapt the model to increase its prediction accuracy on the next input $\mathbf{x}_{t+1}$. The true labels are assumed to come from some task distribution $P(Y_t|X_t, T_t)$, where $T_t$ is the task at time $t$. The tasks themselves change over time, resulting in a non-stationary task distribution, and the identity of the task $T_t$ is unknown to the learner. In real-world settings, tasks might correspond to unknown parameters of the system (e.g., motor malfunction on a robot), user preferences, or other unexpected events. This problem statement covers a range of online learning problems that all require continual adaptation to streaming data and trading off between generalization and specialization.

In our experiments, we use model-based RL as our working example, where the input $\mathbf{x}_t$ is a state-action pair, and the output $\mathbf{y}_t$ is the next state. We discuss this application to model-based RL in Section 6, but we keep the following derivation of our method general for the case of arbitrary online prediction problems.

## 4    ONLINE LEARNING WITH MIXTURE OF META-TRAINED NETWORKS

We discuss our meta-learning for online learning (MOLe) algorithm in two parts: online learning in this section, and meta-learning in the next. In this section, we explain our online learning method that enables effective online learning using a continuous stream of incoming data from a non-stationary task distribution. We aim to retain generalization so as to not lose past knowledge, as well as gain specialization, which is particularly important for learning new tasks that are further out-of-distribution and require more learning. We discuss the process of obtaining a meta-learned prior in Sec. 5, but we first formulate in this section an online adaptation algorithm using SGD with expectation maximization to maintain and adapt a mixture model over task model parameters (i.e., a probabilistic task distribution).

### 4.1    METHOD OVERVIEW

Let $p_{\theta(T_t)}(\mathbf{y}_t|\mathbf{x}_t)$ represent the predictive distribution of our model on input $\mathbf{x}_t$, for an unknown task $T_t$. Our goal is to estimate model parameters $\theta_t(T)$ for each task $T$ in the non-stationary task distribution: This requires inferring the distribution over tasks at each step $P(T_t|\mathbf{x}_t, \mathbf{y}_t)$, using that distribution to make predictions $\hat{\mathbf{y}}_t = p_{\theta(T_t)}(\mathbf{y}_t|\mathbf{x}_t)$, and also using it to update each model from $\theta_t(T)$ to $\theta_{t+1}(T)$. In practice, the parameters $\theta(T)$ of each model will correspond to the weights of a neural network $f_{\theta(T)}$.

Each model begins with some prior parameter vector $\theta^*$, which we will discuss in more detail in Section 5. Since the number of tasks is also unknown, we begin with one task at time step 0, where $\theta_0(T) = \{\theta_0(T_0)\} = \{\theta^*\}$. From here, we continuously update all parameters in $\theta_t(T)$ and add new tasks as needed, in the attempt to model the true underlying process $P(Y_t|X_t, T_t)$. Since task identities are unknown, we must also estimate $P(T_t)$ at each time step. Thus, the online learning problem consists of adapting each $\theta_t(T_i)$ at each time step $t$ according to the inferred task probabilities $P(T_t = T_i|\mathbf{x}_t, \mathbf{y}_t)$. To do this, we adapt the expectation maximization (EM) algorithm and optimize the expected log-likelihood, given by

$$\mathcal{L} = E_{T_t \sim P(T_t|\mathbf{x}_t, \mathbf{y}_t)}[\log p_{\theta_t(T_t)}(\mathbf{y}_t|\mathbf{x}_t, T_t)], \tag{1}$$

where we use $\theta_t(T_t)$ to denote the model parameters corresponding to task $T_t$. Finally, to handle the unknown number of tasks, we employ the Chinese restaurant process to instantiate new tasks as needed.

## 4.2 APPROXIMATE ONLINE INFERENCE

We use expectation maximization (EM) to update the model parameters. In our case, the E step in EM involves estimating the task distribution $P(T_t = T_i|\mathbf{x}_t, \mathbf{y}_t)$ at the current time step, while the M step involves updating all model parameters from $\theta_t(T)$ to obtain the new model parameters $\theta_{t+1}(T)$. The parameters are always updated by one gradient step per time step, according to the inferred task responsibilities.

We first estimate the expectations over all task parameters $T_i$ in the task distribution, where the posterior of each task probability can be written as follows:

$$P(T_t = T_i|\mathbf{x}_t, \mathbf{y}_t) \propto p_{\theta_t(T_i)}(\mathbf{y}_t|\mathbf{x}_t, T_t = T_i)P(T_t = T_i). \tag{2}$$

We then formulate the task prior $P(T_t = T_i)$ using a Chinese restaurant process (CRP) to enable new tasks to be instantiated during a trial. The CRP is an instantiation of a Dirichlet process. In the CRP, at time $t$, the probability of each task $T_i$ should be given by

$$P(T_t = T_i) = \frac{n_{T_i}}{t - 1 + \alpha} \tag{3}$$

where $n_{T_i}$ is the expected number of datapoints in task $T_i$ for all steps $1, \ldots, t-1$, and $\alpha$ is a hyperparameter that controls the instantiation of new tasks. The prior therefore becomes

$$P(T_t = T_i) = \frac{\sum_{t'=1}^{t-1} P(T_{t'} = T_i)}{t - 1 + \alpha} \quad \text{and} \quad P(T_t = T_{\text{new}}) = \frac{\alpha}{t - 1 + \alpha} \tag{4}$$

Combining the prior and likelihood, we derive the following posterior task probability distribution:

$$P(T_t = T_i|\mathbf{x}_t, \mathbf{y}_t) \propto p_{\theta_t(T_i)}(\mathbf{y}_t|\mathbf{x}_t, T_t = T_i) \left[\sum_{t'=1}^{t-1} P(T_{t'} = T_i) + \delta(T_{t'} = T_{\text{new}})\alpha\right] \tag{5}$$

Having estimated the latent task probabilities, we next perform the M step, which improves the expected log-likelihood in Equation 1 based on the inferred task distribution. Since each task starts from the prior $\theta^*$, the values of all parameters in $\theta_t(T)$ after one gradient update are given by

$$\theta_{t+1}(T_i) = \theta^* - \beta \sum_{t'=0}^{t} P_t(T_{t'} = T_i|\mathbf{x}_{t'}, \mathbf{y}_{t'})\nabla_{\theta_{t'}(T_i)} \log p_{\theta_{t'}(T_i)}(\mathbf{y}_{t'}|\mathbf{x}_{t'}) \quad \forall T_i \tag{6}$$

If we assume that all parameters of $\theta_t(T)$ have already been updated for the previous time steps $0, \ldots, t$, we can approximate this update by simply updating all parameters $\theta_t(T)$ on the newest data:

$$\theta_{t+1}(T_i) = \theta_t(T_i) - \beta P_t(T_t = T_i|\mathbf{x}_t, \mathbf{y}_t)\nabla_{\theta_t(T_i)} \log p_{\theta_t(T_i)}(\mathbf{y}_t|\mathbf{x}_t) \quad \forall T_i \tag{7}$$

This procedure is an approximation, since updates to task parameters $\theta_t(T)$ will in reality also change the corresponding task probabilities at previous time steps. However, this approximation removes the need to store previously seen data points and yields a fully online, streaming algorithm. Finally, to fully implement the EM algorithm, we must alternate the E and M steps to convergence at each time step, rolling back the previous gradient update to $\theta_t(T)$ at each iteration. In practice, we found it sufficient to perform the E and M steps only once per time step. While this is a crude simplification, successive time steps in the online learning scenario are likely to be correlated, making this procedure reasonable. However, it is also straightforward to perform multiple steps of EM while still remaining fully online.

We now summarize this full online learning portion of MOLe, and we also outline it in Alg. 1. At the first time step $t = 0$, the task distribution is initialized to contain one entry: $\theta_0(T) = \{\theta_0(T_0)\} = \{\theta^*\}$. At every time step after that, an E step is performed to estimate the task distribution and an M step is performed to update the model parameters. The CRP prior also assigns, at each time step, the probability of adding a new task $T_{\text{new}}$ at the given time step. The parameters $\theta_{t+1}(T_{\text{new}})$ of this new task are adapted from $\theta^*$ on the latest data. The prediction on the next datapoint is then made using the model parameters $\theta_{t+1}(T^*)$ corresponding to the most likely task $T^*$.

---

**Algorithm 1** Online Learning with Mixture of Meta-Trained Networks

---
**Require:** $\theta^*$ from meta-training
  Initialize $t = 0$, $\theta_0(T) = \{\theta_0(T_0)\} = \{\theta^*\}$
  **for** each time step $t$ **do**
    Calculate $p_{\theta_t(T_i)}(\mathbf{y}_t|\mathbf{x}_t, T_t = T_i)$ $\forall T_i$
    Calculate $P_t(T_i) = P_t(T_t = T_i|\mathbf{x}_t, \mathbf{y}_t)$ $\forall T_i$
    Calculate $\theta_{t+1}(T_i)$ by adapting from $\theta_t(T_i)$ $\forall T_i$
    Calculate $\theta_{t+1}(T_{\text{new}})$ by adapting from $\theta^*$
    **if** $P_t(T_{\text{new}}) > P_t(T_i)$ $\forall T_i$ **then**
      Add $\theta_{t+1}(T_{\text{new}})$ to $\theta_{t+1}(T)$
      Recalculate $P_t(T_i)$ using $\theta_{t+1}(T_i)$ $\forall T_i$
      Recalculate $\theta_{t+1}(T_i)$ using updated $P_t(T_i)$ $\forall T_i$
    **end if**
    $T^* = \text{argmax}_{T_i}$ $p_{\theta_{t+1}(T_i)}(\mathbf{y}_t|\mathbf{x}_t, T_t = T_i)$
    Receive next data point $\mathbf{x}_{t+1}$
  **end for**

---

## 5 META-LEARNING THE PRIOR

We formulated an algorithm above for performing online adaptation using continually incoming data. For this method, we choose to meta-train the prior using the model-agnostic meta-learning (MAML) algorithm. This meta-training algorithm is an appropriate choice, because it results in a prior that is specifically intended for gradient-based fine-tuning. Before we further discuss our choice in meta-training procedure, we first give an overview of MAML and meta-learning in general.

Given a distribution of tasks, a meta-learning algorithm produces a learning procedure, which can, in some cases, quickly adapt to a new task. MAML optimizes for an initialization of a deep network that achieves good few-shot task generalization when fine-tuned using a few datapoints from that task. At train time, MAML sees small amounts of data from large numbers of tasks, where data $\mathcal{D}_T$ from each task $T$ can be split into training and validation subsets ($\mathcal{D}_T^{\text{tr}}$ and $\mathcal{D}_T^{\text{val}}$), where $\mathcal{D}_T^{\text{tr}}$ is of size $k$. MAML optimizes for model parameters $\theta$ such that one or more gradients steps on $\mathcal{D}_T^{\text{tr}}$ results in a minimal loss $L$ on $\mathcal{D}_T^{\text{val}}$. In our case, we will set $\mathcal{D}_{T_t}^{\text{tr}} = (\mathbf{x}_t, \mathbf{y}_t)$ and $\mathcal{D}_{T_t}^{\text{val}} = (\mathbf{x}_{t+1}, \mathbf{y}_{t+1})$, and the loss $L$ will correspond to negative log likelihood. A good $\theta$ that allows such adaptation to be successful across various meta-training tasks is thus a good network initialization from which adaptation can solve various new tasks that are related to the previously seen tasks. The MAML objective is defined as follows:

$$\min_\theta \sum_T L(\theta - \eta\nabla_\theta L(\theta, \mathcal{D}_T^{\text{tr}}), \mathcal{D}_T^{\text{val}}) = \min_\theta \sum_T L(\phi_T, \mathcal{D}_T^{\text{val}}). \tag{8}$$

Here, $\eta$ is the inner learning rate. Once this meta-objective is optimized, the resulting $\theta^*$ acts as a prior from which fine-tuning can occur at test-time, using recent experience from $\mathcal{D}_{T_{\text{test}}}^{\text{tr}}$ as follows:

$$\phi_{T_{\text{test}}} = \theta^* - \eta\nabla_{\theta^*} L(\theta^*, \mathcal{D}_{T_{\text{test}}}^{\text{tr}}). \tag{9}$$

Here, $\phi_{T_{\text{test}}}$ is adapted from the meta-learned prior $\theta^*$ to be more representative for the current time.

Although Finn et al. (2017) demonstrated this fast adaptation of deep neural networks and Nagabandi et al. (2018) extended this framework to model-based meta RL, these methods address adaptation in the $k$-shot setting, always adapting directly from the meta-learned prior and not allowing further adaptation or specialization. In this work, we have extended these capabilities by enabling more evolution of knowledge through a temporally-extended online adaptation procedure.

While our procedure for continual online learning is still initialized with this meta-training for $k$-shot adaptation (i.e., MAML), we found that this prior was sufficient to enable effective continual online adaptation at test time. The intuitive rationale for this is that MAML trains the model to be able to change significantly using only a small number of datapoints and gradient steps. Note that this meta-trained prior can be used at test time in (a) a $k$-shot setting, similar to how it was trained, or it can be used at test time by (b) taking substantially more gradient steps away from this prior. We show in Sec. 7 that our method outperforms both of these methods, but the mere ability to use this meta-learned prior in these ways makes the use of MAML enticing.

We note that it is quite possible to modify the MAML algorithm to optimize the model directly with respect to the weighted updates discussed in Section 4.2. This simply requires computing the task weights (the E step) on each batch during meta-training, and then constructing a computation graph where all gradient updates are multiplied by their respective weights. Standard automatic differentiation software can then compute the corresponding meta-gradient. For short trial lengths, this is not substantially more complex than standard MAML; for longer trial lengths, truncated backpropagation is an option. Although such a meta-training procedure better matches the way that the model is used during online adaptation, we found that it did not substantially improve our results. While it's possible that the difference might be more significant if meta-training for longer-term adaptation, this observation does suggest that simply meta-training with MAML is sufficient for enabling effective continuous online adaptation in non-stationary multi-task settings. To clarify, although this modified training procedure of incorporating the EM weight updates (during meta-training) did not explicitly improve our results, we see that test-time performance did indeed improve with using more data for the standard MAML meta-training procedure (see Appendix).

## 6 APPLICATION TO MODEL-BASED RL

In our experiments, we apply MOLe to model-based reinforcement learning. RL in general aims to act in a way that maximizes the sum of future rewards. At each time step $t$, the agent executes action $\mathbf{a}_t \in A$ from state $\mathbf{s}_t \in S$, transitions to the next state $\mathbf{s}_{t+1}$ according the transition probabilities (i.e., dynamics) $p(\mathbf{s}_{t+1}|\mathbf{s}_t, \mathbf{a}_t)$ and receives rewards $r_t = r(\mathbf{s}_t, \mathbf{a}_t)$. The goal at each step is to execute the action $\mathbf{a}_t$ that maximizes the discounted sum of future rewards $\sum_{t'=t}^{\infty} \gamma^{t'-t} r(\mathbf{s}_{t'}, \mathbf{a}_{t'})$, where discount factor $\gamma \in [0, 1]$ prioritizes near-term rewards. In model-based RL, in particular, the predictions from a known or learned dynamics model are used to either learn a policy, or are used directly inside a planning algorithm to select actions that maximize reward.

In our work, the underlying distribution that we aim to model is the dynamics distribution $p(\mathbf{s}_{t+1}|\mathbf{s}_t, \mathbf{a}_t, T_t)$, where the unknown $T_t$ represents the underlying settings (e.g., state of the system, external details, environmental perturbations, etc.). The goal for MOLe is to estimate this distribution with a predictive model $p_\theta$. To instantiate MOLe in this context of model-based RL, we follow Algorithm 1 with the following specifications:

(1) We set the input $\mathbf{x}_t$ to be the concatenation of $K$ previous states and actions, given by $\mathbf{x}_t = [\mathbf{s}_{t-K}, \mathbf{a}_{t-K}, \ldots, \mathbf{s}_{t-2}, \mathbf{a}_{t-2}, \mathbf{s}_{t-1}, \mathbf{a}_{t-1}]$, and the output to be the corresponding next states $\mathbf{y}_t = [\mathbf{s}_{t-K+1}, \ldots, \mathbf{s}_{t-1}, \mathbf{s}_t]$. This provides us with a slightly larger batch of data for each online update, as compared to using only the data from the given time step. Since individual time steps at high frequency can be very noisy, using the past $K$ transitions helps to damp out the updates.

(2) The predictive model $p_\theta$ represents each of these underlying transitions as an independent Gaussian such that $p_\theta(\mathbf{y}_t|\mathbf{x}_t) = \prod_{t'=t-K}^{t-1} p(\mathbf{s}_{t'+1}|\mathbf{s}_{t'}, \mathbf{a}_{t'})$, where each $p(\mathbf{s}_{t+1}|\mathbf{s}_t, \mathbf{a}_t)$ is parameterized with a Gaussian given by mean $f_\theta(\mathbf{s}_t, \mathbf{a}_t)$ and constant variance $\sigma^2$. We implement this mean dynamics function $f_\theta(s_t, a_t)$ as a neural network model with three hidden layers each of dimension 500, and ReLU nonlinearities.

(3) To calculate the new task parameter $\theta_{t+1}(T_{\text{new}})$, which may or may not be added to the task distribution $\theta_{t+1}(T)$, we use a set of $K$ nearby datapoints that is separate from the set $\mathbf{x}_t$. This is done to avoid calculating the parameter using the same dataset on which it is evaluated, since $P_t(T_{\text{new}})$ comes from evaluating the parameter on the data $\mathbf{x}_t$.

(4) Unlike standard online streaming tasks where the next data point $\mathbf{x}_{t+1}$ is just given, the incoming data point (i.e., the next visited state) in this case is influenced by the predictive model itself. This is because, after the most likely task $T^*$ is selected from the possible tasks, the predictions from the model $p_{\theta_{t+1}(T^*)}$ are used by the controller to plan over a sequence of future actions $\mathbf{a}_0, \ldots, \mathbf{a}_H$ and select the actions that maximize future reward. Note that the planning procedure is based on stochastic optimization, following prior work (Nagabandi et al., 2018), and we provide more details in the appendix. Since the controller's action choice determines the next data point, and since the controller's choice is dependent on the estimated model parameters, it is even more crucial in this setting to appropriately adapt the model.

5) Finally, note that we attain $\theta^*$ from meta-training using model-agnostic meta-learning (MAML), as mentioned in the method above. However, in this case, MAML is performed in the loop of model-based RL. In other words, the model parameters at a given iteration of meta-training are used by the controller to generate on-policy rollouts, the data from these rollouts is then added to the dataset for MAML, and this process repeats until the end of meta-training.

# 7 EXPERIMENTS

The questions that we aimed to study from our experiments include: Can MOLe 1) autonomously discover some task structure amid a stream of non-stationary data? 2) adapt to tasks that are further outside of the task distribution than can be handled by a $k$-shot learning approach? 3) recognize and revert to tasks it has seen before? 4) avoid overfitting to a recent task to prevent deterioration of performance upon the next task switch? 5) outperform other methods?

To study these questions, we conduct experiments on agents in the MuJoCo physics engine (Todorov et al., 2012). The agents we used are a half-cheetah (S R21, A R6) and a hexapedal crawler (S R50, A R12). Using these agents, we design a number of challenging online learning problems that involve multiple sudden and gradual changes in the underlying task distribution, including tasks that are extrapolated from those seen previously, where online learning is criticial. Through these experiments, we aim to build problem settings that are representative of the types of disturbances and shifts that a real RL agent might encounter.

We present results and analysis of our findings in the following three sections, and videos can be found at `https://sites.google.com/berkeley.edu/onlineviameta`. In our experiments, we compare to several alternative methods, including two approaches that leverage meta-training and two approaches that do not:

(a) **k-shot adaptation with meta-learning**: Always adapt from the meta-trained prior $\theta^*$, as typically done with meta-learning methods (Nagabandi et al., 2018). This method is often insufficient for adapting to tasks that are further out of distribution, and the adaptation is also not carried forward in time for future use.

(b) **continued adaptation with meta-learning**: Always take gradient steps from the previous time step's parameters. This method oftens overfits to recently observed tasks, so it should indicate the importance of our method effectively identifying task structure to avoid overfitting and enable recall.

(c) **model-based RL**: Train a model on the same data as the methods above, using standard supervised learning, and keep this model fixed throughout the trials (i.e., no meta-learning and no adaptation).

(d) **model-based RL with online gradient updates**: Use the same model from model-based RL (i.e., no meta-learning), but adapt it online using gradient-descent at run time. This is representative of commonly used dynamic evaluation methods (Rei, 2015; Krause et al., 2017; 2016; Fortunato et al., 2017).

## 7.1 TERRAIN SLOPES ON HALF-CHEETAH

We start with the task of a half-cheetah (Fig. 1) agent, traversing terrains of differing slopes. The prior model is meta-trained on data from terrains with random slopes of low magnitudes, and the test trials are executed on difficult out-of-distribution tasks such as basins, steep hills, etc. As shown in Fig. 2, neither model-based RL nor model-based RL with online gradient updates perform well on these out-of-distribution tasks, even though those models were trained on the same data that the meta-trained model received. The bad performance of the model-based RL approach indicates the need for model

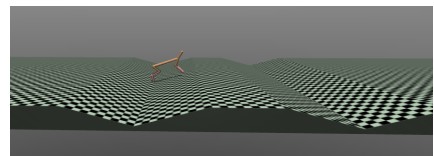

Figure 1: Half-cheetah agent, shown traversing a landscape with 'basins'

adaptation (as opposed to assuming a single model can encompass everything), while the bad performance of model-based RL with online gradient updates indicates the need for a meta-learned initialization to enable online learning with neural networks.

For the three meta-learning and adaptation methods, we expect continued adaptation with meta-learning to perform poorly due to continuous gradient steps causing it to overfit to recent data; that is, we expect that experience on the upward slopes to lead to deterioration of performance on downward slopes, or something similar. However, based on both our qualitative and quantitative results, we see that the meta-learning procedure seems to have initialized the agent with a parameter space in which these various "tasks" are not seen as substantially different, where online learning by SGD performs well. This suggests that the meta-learning process finds a task space where there is an easy skill transfer between slopes; thus, even when MOLe is faced with the option of switching tasks or adding new tasks to its dynamic latent task distribution, it chooses not to do so (Fig. 3). Unlike findings that we will see later, it is interesting that the discovered task space here does not correspond to human-distinguishable categorical labels. Finally, note that these tasks of changing slopes are not particularly similar to each other (and that the discovered task space is indeed useful), because the two non-meta-learning baselines do indeed fail on these test tasks despite having similar training performance on the shallow training slopes.

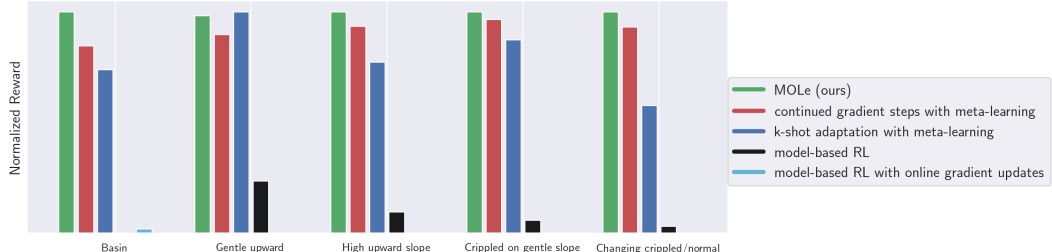

Figure 2: Results on half-cheetah terrain traversal. The poorly performing model-based RL shows that a single model is not sufficient, and model-based RL with online gradient updates shows that a meta-learned initialization is critical. The three meta-learning approaches perform similarly; however, the performance of k-shot adaptation deteriorates when the task calls for taking multiple gradient steps away from the prior.

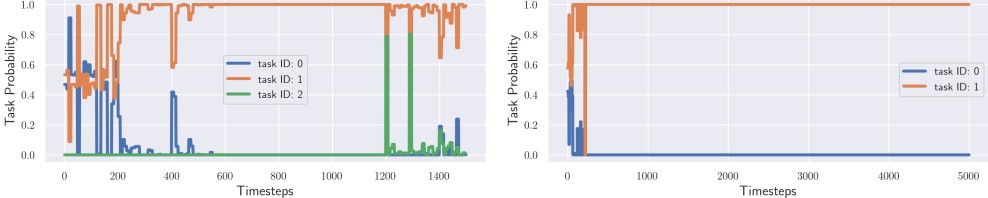

Figure 3: Latent task distribution over time for two half-cheetah landscape traversal tasks, where encountered terrain slopes vary within each run. Interestingly, we find that MOLe chooses to only use a single latent task variable to describe varying terrain.

## 7.2 HALF-CHEETAH MOTOR MALFUNCTIONS

While the findings from the half-cheetah on sloped terrains illustrate that separate task parameters aren't always necessary for what might externally seem like separate tasks, we also want to study agents that experience more drastically-changing non-stationary task distributions during their experience in the world. For this set of experiments, we train all models on data where a single actuator is selected at random to experience a malfunction during the rollout. In this case, malfunction means that the polarity or magnitude of actions applied to that actuator are altered. Fig. 4 shows the results of various methods on drastically out-of-distribution test tasks, such as altering all actuators at once. The left of Fig. 4 shows that when the task distribution during the test trials contains only a single task, such as 'sign negative' where all actuators are prescribed to be the opposite polarity, then continued adaptation performs well by repeatedly performing gradient updates on incoming data. However, as shown in the other tasks of Fig. 4, the performance of this continue adaptation substantially deteriorates when the agent experiences a non-stationary task distribution. Due to over-specialization on recent incoming data, such methods that continuously adapt tend to forget and lose previously existing skills. This overfitting and forgetting of past skills is also illustrated in the consistent performance deterioration shown in Fig. 4. MOLe, on the other hand, dynamically builds a probabilistic task distribution and allows adaptation to these difficult tasks, without forgetting past skills. We show a sample task setup in Fig. 5, where the agent experiences alternating periods of

normal and crippled-leg operation. This plot shows the successful recognition of new tasks as well as old tasks; note that both the recognition and adaptation are all done online, without using a bank of past data to perform the adaptation, and without a human-specified set of task categories.

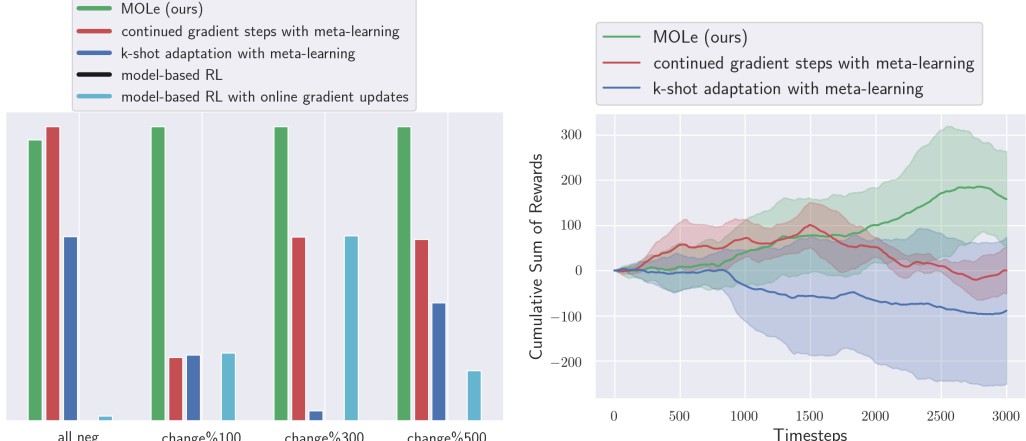

Figure 4: Results on the motor malfunction trials, where different trials are shown task distributions that modulate at different frequencies (or stay constant, for the first category). Here, online learning is critical for good performance, k-shot adaptation is insufficient for such different tasks, and continued gradient steps leads to overfitting to recently seen data.

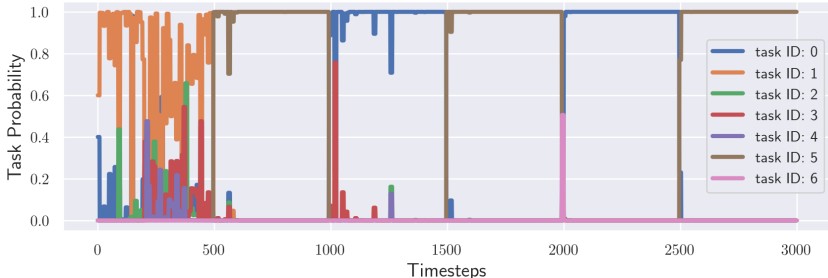

Figure 5: Latent task variable distribution over the course of an online learning trial where the underlying motor malfunction changes every 500 timesteps. We find that MOLe is able to successfully recover the task structure, recognize when the underlying task has changed, and recall previously seen tasks.

## 7.3 CRIPPLING OF END EFFECTORS ON SIX-LEGGED CRAWLER

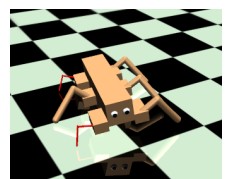

To further examine the effects of our continual online adaptation algorithm, we study another, more complex agent: a 6-legged crawler (Fig. 6). In these experiments, models are trained on random joints being crippled (i.e., unable to apply actuator commands). In Fig. 7, we present two illustrative test tasks: (1) the agent sees a set configuration of crippling for the duration of its test-time experience, and (2) the agent receives alternating periods of experience, between regions of normal operation and regions of having crippled legs.

Figure 6: Six-legged crawler robot, shown with crippled legs.

The first setting is similar to data seen during training, and thus, we see that even the model-based RL and model based-RL with online gradient updates baselines do not fail. The methods that include both meta-learning and adaptation, however, do have higher performance. Furthermore, we see again that continued gradient steps in this case of a single-task setting is not detrimental. The second setting's non-stationary task distribution (when the leg crippling is dynamic) illustrates the need for online adaptation (model-based RL fails), the need for a good prior to adapt from (failure of model-based RL with online gradient updates), the harm of overfitting to recent experience and thus forgetting older skills (low performance of continued gradient steps), and the need for further adaptation away from the prior (limited performance of k-shot adaptation). With MOLe, this agent is able to build its own representation of "task" switches, and we see that this switch does indeed correspond to recognizing regions

of leg crippling (left of Fig. 7.3). The plot of the cumulative sum of rewards (right of Fig. 7.3) of each of the three meta-learning plus adaptation methods includes this same task switch pattern every 500 steps: Here, we can clearly see that steps 500-1000 and 1500-2000 were the crippled regions. Continued gradient steps actually performs worse on the second and third times it sees normal operation, whereas MOLe is noticeably better as it sees the task more often. Note that this improvement of both skills is possible with MOLe, where development of one skill actually does not hinder the other.

Finally, we examine experiments where the crawler experiences (during each trial) walking straight, making turns, and sometimes having a crippled leg. The performance during the first 500 time steps of "walking forward in a normal configuration" for continued gradient steps was comparable to MOLe (+/-10% difference), but its performance during the last 500 time steps of "walking forward in a normal configuration" was **200%** lower. Note this detrimental effect of performing updates without allowing for separate task specialization/adaptation.

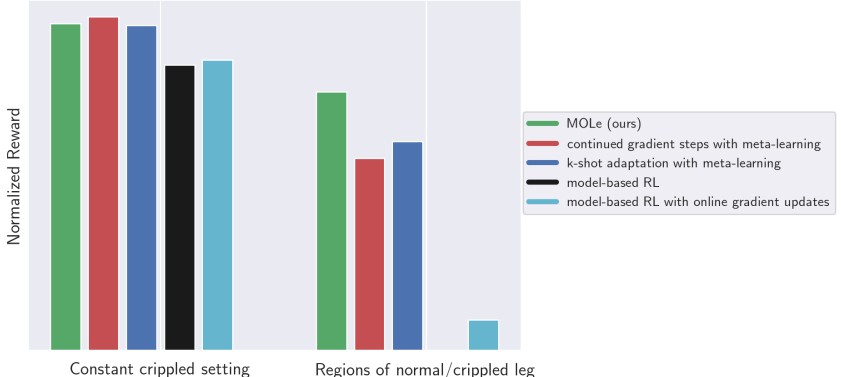

Figure 7: Quantitative results on crawler. For a fixed task, adaptation is not necessary and all methods perform well. In contrast, when tasks change dynamically within the trial, only MOLe effectively learns online.

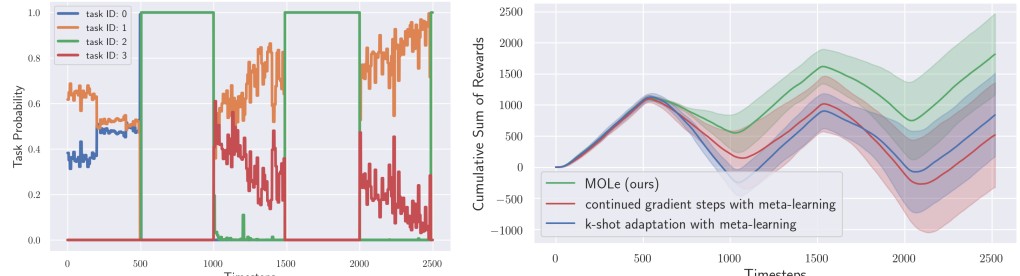

Figure 8: Results on crawler experiments. Left: Online recognition of latent task probabilities for alternating periods of normal/crippled experience. Right: MOLe improves from seeing the same tasks multiple times.

## 8 DISCUSSION

We presented an online learning method for neural network models that can handle non-stationary, multi-task settings within each trial. Our method adapts the model directly with SGD, where an EM algorithm uses a Chinese restaurant process prior to maintain a distribution over tasks and handle non-stationarity. Although SGD generally makes for a poor online learning algorithm in the streaming setting for large parametric models such as deep neural networks, we observe that, by (1) meta-training the model for fast adaptation with MAML and (2) employing our online algorithm for probabilistic updates at test time, we can enable effective online learning with neural networks. In our experiments, we applied this approach to model-based RL, and we demonstrated that it could be used to adapt the behavior of simulated robots faced with various new and unexpected tasks. Our results showed that our method can develop its own notion of task, continuously adapt away from the prior as necessary (to learn even tasks that require more adaptation), and recall tasks it has seen before. While we use model-based RL as our evaluation domain, our method is general and could be applied to other streaming and online learning settings. An exciting direction for future work would be to apply our method to domains such as time series modeling and active online learning.

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

## A   TEST-TIME PERFORMANCE VS TRAINING DATA

We verify below that as the meta-trained models are trained with more data, their performance on test tasks does improve.

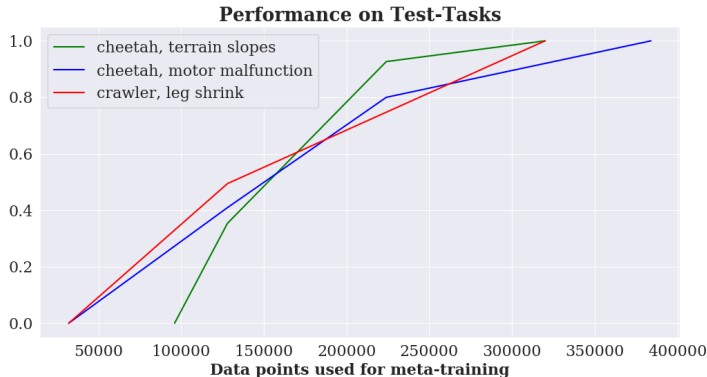

Figure 9: Performance on test tasks (i.e., unseen during training) of models that are meta-trained with differing amounts of data. Performance numbers here are normalized per agent, between 0 and 1.

## B   HYPERPARAMETERS

In all experiments, we use a dynamics model consisting of three hidden layers, each of dimension 500, with ReLU nonlinearities. The control method that we use is random-shooting model predictive control (MPC) where 1000 candidate action sequences each of horizon length H=10 are sampled at each time step, fed through the predictive model, and ranked by their expected reward. The first action step from the highest-scoring candidate action sequence is then executed before the entire planning process repeats again at the next time step.

Below, we list relevant training and testing parameters for the various methods used in our experiments. # Task/itr corresponds to the number of tasks sampled during each iteration of collecting data to train the model, and # TS/itr is the total number of times steps collected during that iteration (sum over all tasks).

Table 1: Hyperparameters for train-time

|  | Iters | Epochs | # Tasks/itr | # TS/itr | K | outer LR | inner LR ($\eta$) |
|---|---|---|---|---|---|---|---|
| **Meta-learned approaches (3)** | 12 | 50 | 16 | 2000-3000 | 16 | 0.001 | 0.01 |
| **Non-meta-learned approaches (2)** | 12 | 50 | 16 | 2000-3000 | 16 | 0.001 | N/A |

Table 2: Hyperparameters for run-time

|  | $\alpha$ (CRP) | LR (model update) | K (previous data) |
|---|---|---|---|
| **MOLe (ours)** | 1 | 0.01 | 16 |
| **continued adaptation with meta-learning** | N/A | 0.01 | 16 |
| **k-shot adaptation with meta-learning** | N/A | 0.01 | 16 |
| **model-based RL** | N/A | N/A | N/A |
| **model-based RL with online gradient updates** | N/A | 0.01 | 16 |

## C   CONTROLLER

As mentioned in Section 6, we use the learned dynamics model in conjunction with a controller to select the next action to execute. The controller uses the learned model $f_\theta$ together with a reward function $r(\mathbf{s}_t, \mathbf{a}_t)$ that encodes the desired task. Many methods could be used to perform this action selection, including cross entropy method (CEM) (Botev et al., 2013) or model predictive path integral control (MPPI) (Williams et al., 2015), but in our experiments, we use a random-sampling shooting method Rao (2009).

At each time step $t$, we randomly generate $N$ candidate action sequences with $H$ actions in each sequence.

$$A_t^i = \{\mathbf{a}_t^i, \dots, \mathbf{a}_{t+H}^i\} \tag{10}$$

We then use the learned dynamics model to predict the resulting states of executing these candidate action sequences.

$$S^i = \{\hat{\mathbf{s}}_{t+1}^i, \dots, \hat{\mathbf{s}}_{t+H+1}^i\} \ \text{ where } \ \hat{\mathbf{s}}_{p+1}^i = f_\theta(\hat{\mathbf{s}}_p^i, \mathbf{a}_p^i) \tag{11}$$

Next, we use the reward function to select the action sequence with the highest associated predicted reward.

$$i^* = \text{argmax}_i \sum_{t'=t}^{t'=t+H} r(\hat{\mathbf{s}}_{t'}^i, \mathbf{a}_{t'}^i) \tag{12}$$

Next, rather than executing the entire sequence $A_t^{i^*}$ of selected optimal actions, we use a model predictive control (MPC) framework to execute only the first action $\mathbf{a}_t^{i^*}$ from the current state $\mathbf{s}_t$. We then replan at the next time step; This use of MPC can compensate for model inaccuracies by preventing accumulating errors, since we replan at each time step using updated state information.

