# OpenReview forum: "Deep Online Learning Via Meta-Learning: Continual Adaptation for Model-Based RL"
_ICLR.cc/2019/Conference_

### Official Review · AnonReviewer1 · 2018-10-31
**Useful method for online adaptation to sudden changes in the modeled environment**

**Rating:** 7
**Confidence:** 3

**Review:**

The paper introduces a method for online adaptation of a model that is expected to adapt to changes in the environment the model models. The method is based on a mixture model, where new models are spawned using a Chinese restaurant process, and where each newly spawned model starts with weights that have been trained using meta-learning to quickly adapt to new dynamics. The method is demonstrated on model-based RL for a few simple benchmarks.

The proposed method is well justified, clearly presented, and the experimental results are convincing. The paper is generally clear and well written. The method is clearly most useful for situations where the environment suddenly changes, which is relevant in some real-world problems. As a drawback, using a mixture model (that also grows with time) for such modelling can be considered quite heavy in some situations. Nevertheless, the idea of combining a spawning process with meta-learned priors is neat, and clearly works well.

Minor comments:
- Algorithm 1: is the inequality correct, and is T* supposed to be an argmin instead of argmax?

---

> ### Author Response · Authors · 2018-11-22
> **Thank you for your review!**
>
> Thank you for your review. We have corrected the typo in both places of Algorithm 1: it should indeed have been the opposite inequality sign, and argmin instead of argmax.
>
> We definitely agree with your comment that a mixture model that grows with time can sometimes be considered quite heavyweight. This is precisely where we plan to focus the efforts of our future work, by introducing a refreshing scheme where an offline retraining step can periodically condense the mixture model into fewer components (perhaps in a batch-mode training setting, so not all past data needs to be saved). We are also interested in goals such as making this mixture only as big as the agent “needs” it to be, allowing for better and more compressed sharing and organization of seen data. The performance of this current method makes us hopeful and excited to work toward such future work in this area.

---

### Official Review · AnonReviewer2 · 2018-11-03
**This was a nice proposal of a nonparametric mixture model of NNs initialized with meta-learning for supervised learning under nonstationary distributions.**

**Rating:** 7
**Confidence:** 3

**Review:**

The paper presents a nonparametric mixture model of neural networks for learning in an environment with a nonstationary distribution. The problem setup includes having access to only a few "modes" of the distribution. Training of the initial model occurs with MAML, and distributional changes during test/operation are handled by a combination of online adaptation and creations of new mixture components when necessary. The mixture is nonparametric and modeled with a CRP. The application considered in the paper is RL, and the experiments compare proposed model against baselines that do not utilize meta-learning (achieved in the proposed method with MAML), and baselines which utilize only a single model component.

I thought the combination of meta-learning and a CRP was a neat way to tackle the problem of modeling and learning the "modes" of a nonstationary distribution. Applications in other domains would have been nice, but the presented results in RL sufficiently demonstrate the benefits of the proposed method.

* Questions/Comments

Figure 3 left vs right?

Is the test in the middle of Algorithm 1 correct?

---

> ### Author Response · Authors · 2018-11-22
> **Thank you for your review!**
>
> Thank you for your review. We have corrected the typo in the test in the middle of Algorithm 1: it should have been argmin instead of argmax. We have also clarified the caption of figure 3 to indicate that the two plots simply illustrate two different runs for the indicated agent, showing that our method chooses to assign only a single task variable even throughout runs including changing terrain slopes.

---

### Official Review · AnonReviewer3 · 2018-11-04
**Nice work**

**Rating:** 7
**Confidence:** 3

**Review:**

The authors proposed a new method to learn streaming online updates for neural networks with meta-learning and applied it to multi-task reinforcement learning. Model-agnostic meta-learning is used to learn the initial weight and task distribution is learned with the Chinese restaurant process. It sounds like an interesting idea and practical for RL. Extensive experiments show the effectiveness of the proposed method.

The authors said that online updating the meta-learner did not improve the results, which is a bit surprised. Also how many data are meta-trained is not clearly described in the paper. Maybe the authors can compare the results with less data for meta-training.

---

> ### Author Response · Authors · 2018-11-22
> **Thank you for your review!**
>
> Thank you for your review. We added an appendix to the paper that addresses your question, and we have also added this information (as well as illustrative videos) to the project website. To illustrate results with less meta-training data, we have evaluated the test-time performance of models from various meta-training iterations, showing that performance does indeed improve with more meta-training data. To clarify, this statement of performance improving with meta-training data is different from the statement in the text regarding online updating the meta-learner not improving results. We meant that incorporating the EM weight updates during meta-training did not improve results, but we did not mean that additional meta-learning was harmful. We added text at the end of section 5 in the updated paper to reduce the potential for confusion.
>
> Regarding the amount of data used, the number of datapoints used during metatraining on each of the agents in our experiments is 382,000: This is 12 iterations of alternating model training plus on-policy rollouts, where each iteration collects data from 16 different environment settings, and each setting consists of 2000 datapoints. At a simulator timestep of 0.02sec/step, this sample complexity converts to around only 2 hours of real-world data.

---

### Meta-Review · Area_Chair1 · 2018-12-14
**Solid contribution, relevant to some interesting real world settings**

**Confidence:** 4
**Recommendation:** Accept (Poster)

**Metareview:**

The reviewers appreciated this contribution, particularly its ability to tackle nonstationary domains which are common in real-world tasks.